# Developing a core outcome set for traumatic brachial plexus injuries: a systematic review of outcomes

Caroline Miller ,[1,2] Jane Cross,[1] Joel O'Sullivan,[2] Dominic M Power,[3] Derek Kyte,[4] Christina Jerosch-Herold [1]

[1]School of Health Sciences, The Queens Building, University of East Anglia, Norwich, UK
[2]Physiotherapy Department, Queen Elizabeth Hospital Birmingham, Birmingham, UK
[3]The Birmingham Peripheral Nerve Injury Service, Queen Elizabeth Hospital Birmingham, Birmingham, UK
[4]Institute of Applied Health Research, University of Birmingham, Birmingham, UK

**Correspondence to**
Caroline Miller;
caroline.miller@uea.ac.uk

## ABSTRACT

**Objective** To identify what outcomes have been assessed in traumatic brachial plexus injury (TBPI) research to inform the development of a core outcome set for TBPI.

**Design** Systematic review.

**Method** Medline (OVID), EMBASE, CINAHL and AMED were systematically searched for studies evaluating the clinical effectiveness of interventions in adult TBPIs from January 2013 to September 2018 updated in May 2021. Two authors independently screened papers. Outcome reporting bias was assessed. All outcomes were extracted verbatim from studies. Patient-reported outcomes or performance outcome measures were extracted directly from the instrument. Variation in outcome reporting was determined by assessing the number of unique outcomes reported across all included studies. Outcomes were categorised into domains using a prespecified taxonomy.

**Results** Verbatim outcomes (n=1491) were extracted from 138 studies including 32 questionnaires. Unique outcomes (n=157) were structured into 4 core areas and 11 domains. Outcomes within the musculoskeletal domain were measured in 86% of studies, physical functioning in 25%, emotional functioning in 25% and adverse events in 33%. We identified 63 different methods for measuring muscle strength, 16 studies for range of movement and 63 studies did not define how they measured movement. More than two-thirds of the outcomes were incompletely reported in prospective studies.

**Conclusion** This review of outcome reporting in TBPI research demonstrated an impairment focus and heterogeneity. A core outcome set would ensure standardised and relevant outcomes are reported to facilitate future systematic review and meta-analysis.

**PROSPERO registration number** CRD42018109843.

## INTRODUCTION

A traumatic brachial plexus injury (TBPI) is a major injury to the brachial plexus. It can result in significant functional, social, psychological and economic effects,[1 2] with most occurring in young men as a result of motorbike accidents.[3] Survival from major trauma is increasing,[4] and with this an increase in the incidence of TBPI,[5] which accounts for 1.2% of polytrauma.[6] The complex and chronic nature of the injury is associated with significant healthcare costs,[7] in addition to indirect costs estimated at up to US$2.34 million (in 2017 US dollars) over the lifetime of a manual labourer in the USA with a TBPI.[8] There are multiple strategies for managing a patient with a TBPI with recent advancements in nerve microsurgery[9] and robotics,[10] resulting in increased treatment options. The choice of treatment should be made using up-to-date, high-quality scientific evidence.[11 12]

Ideally, a meta-analysis would identify the most effective treatment for an individual with a TBPI, however, such analysis requires homogenous outcome measurement and reporting across studies to enable optimum synthesis. Indeed, despite increasing numbers of TBPI studies, outcome heterogeneity and poorly defined outcomes have been highlighted as a significant challenge to evidence synthesis in two recent systematic reviews.[13 14] There is now an international agreement that the definition of a core outcome set (COS) for TBPI is a priority.[15 16] A COS is a minimum agreed set of outcomes to be reported and measured in all studies.[17 18] Development of a COS has been shown to reduce heterogeneity of outcome reporting in other health conditions, with 81% of trialists in rheumatoid arthritis (RA) now measuring the COS for RA.[19]

## Strengths and limitations of this study

► This study is a comprehensive and systematic review of clinical outcomes reported in the traumatic brachial plexus studies from 2013 to 2021 inclusive.
► Unique outcomes were systematically categorised into a clear taxonomy to inform the development of a core outcome set.
► Definition of unique outcomes and categorisation were conducted by researchers and clinicians to account for multidisciplinary perspectives.
► Outcome reporting bias was assessed in included prospective and randomised controlled trials.

To date a minimum set of outcomes, important to patients and professionals for reporting in TBPI studies, has not been agreed. The choice of what are important outcomes to measure in TBPI is complex due to patient heterogeneity with different mechanisms, locations and severity of injury. COS methodology is continuously being refined and promoted by the Core Outcome Measures in Effectiveness Trials (COMET) initiative.[20] Development of a COS usually begins with identification of a long list of outcomes which is then prioritised through a consensus process. This systematic review sits within the larger global COMBINE (Core Outcome Measures in Brachial plexus INjuriEs) project to identify a COS for TBPI. A Delphi study and consensus meeting, informed by data from this systematic review and interviews with people with the injury, will prioritise the final COS for TBPI.

As a first step in the development of an international COS for TBPI, we conducted a systematic review to identify outcomes reported and measurement instruments used and their timing in the literature. The final step of the global project will match the COS to existing validated measurement instruments and make recommendations on when they should be collected, therefore it was necessary to identify currently used instruments and their timepoints also.

The aims of this review were as follows:
1. Identify what outcome domains are assessed in studies evaluating surgical and non-surgical treatment for TBPI.
2. Compare the definitions of outcomes and timepoints of outcomes assessed.
3. Assess selective reporting bias in included prospective studies and randomised controlled trials.
4. Identify how the outcomes were measured, that is, what validated or non-validated instruments are used.

## METHODS
We followed the methods described in the Cochrane Handbook for Systematic Reviews of Interventions,[21] and report in accordance with the Preferred Reporting Items for Systematic Reviews and Meta-Analysis (PRISMA) guidelines.[22] Deviations from the protocol are reported in online supplemental file 1.

### Identification of studies
We conducted an electronic search of Medline (OVID), EMBASE (OVID), CINAHL and AMED on 18 September 2018. Studies published between 1 January 2013 and 18 September 2018 were included to reflect outcomes employed in current TBPI care. An example of the search strategy for Ovid MEDLINE is presented in online supplemental file 2. The thesaurus vocabulary of each database was used to adapt search terms. Boolean operators (AND, OR) were used to narrow or widen the search and no language restrictions were applied. The search was re-run on 7 May 2021 to identify any additional outcomes.

### Study eligibility
Studies were included if they met the following criteria:

#### Study type
Any controlled and uncontrolled experimental and observational studies evaluating interventions in TBPI including case reports, case series, case studies, prospective and retrospective cohort studies, randomised and non-randomised clinical trials. When the search was re-run in May 2021, only prospective cohort and clinical trials were included. We excluded conference proceedings, abstract only publications and those not involving human subjects.

#### Participants
Studies reporting outcomes in individuals with TBPI aged ≥16 years. Studies of patients with obstetric brachial plexus injuries were excluded.

#### Interventions
Any surgical or non-surgical intervention for TBPI.

#### Outcomes
All outcomes reported in the published abstract, methods or results. These included physiological and functional outcomes, adverse events and patient-reported outcomes (PROs) either reported in the study or subsequently extrapolated from the PRO instruments.

#### Language
Non-English language publications were included.

### Study selection process
The reference management software Mendeley was used to compile the literature, with duplicates removed. Authors (CM and JOS) independently screened the titles and then the abstracts against the eligibility criteria. Disagreements were discussed and a third reviewer (CJH) was involved when required. Studies appearing to meet the inclusion criteria based on title and abstract were retrieved as full-text articles, and were read to assess for eligibility with decisions on inclusion and exclusion recorded. Disagreements in study selection were resolved by discussion within the research team (CM, JCand CJH).

### Quality assessment
The aim of this review was to identify outcomes reported in studies rather than synthesise data on intervention effectiveness. However, selective outcome reporting can provide information on what outcomes authors prioritise. We used a modified version of the matrix by Kirkham *et al*[23 24] to assess outcome reporting bias (ORB) in prospective studies and randomised controlled trials (see ORB instrument in online supplemental file 3). Two independent reviewers (CM and JOS) performed the assessment of ORB for all outcomes.

### Data extraction
Data were extracted into a piloted data extraction sheet (Microsoft Excel). General data extracted from each

study included author, study design, recruiting country, publication year, number of participants, gender, mean age, level of TBPI and intervention tested. The following information was extracted regarding outcomes: each outcome reported (verbatim), area of body assessed if relevant (shoulder, elbow, wrist or hand), method of administration, name of measure, timepoints of measure and reported complications. The number of outcomes per study was also documented.

Data extraction was performed independently by CM and JOS for the first 20% of included studies. These were compared, and disagreements discussed and resolved through debate or discussion with a third reviewer (CJH). Following this a further 10% of studies had data extracted by both CM and JOS. Owing to the high level of agreement between reviewers (91% agreement) on outcomes extracted, at this stage, the remaining studies underwent extraction by a single reviewer (CM).

Where a validated PRO or performance outcome measurement was used and composed of multiple items,

the following data were extracted by the first author: verbatim name of the instrument, verbatim wording for each individual item. A performance outcome measurement was defined as 'A measurement based on a standardized task performed by a patient that is administered and evaluated by an appropriately trained individual or is independently completed'.[25] The frequency of use of instruments was noted and compared between studies. The instruments were categorised as: (1) general health (generic—for use with any patient); (2) upper limb physical function (region-specific); (3) symptom or domain specific (to assess a single symptom, eg, pain) and (4) condition specific. Timepoints of measurement of all outcomes were noted. If the outcome was assessed at different timepoints, then all timings were recorded.

## Classification of outcomes into domains and defining unique outcomes

Identically worded and spelled verbatim outcomes were removed at this stage. Identical outcomes measured over

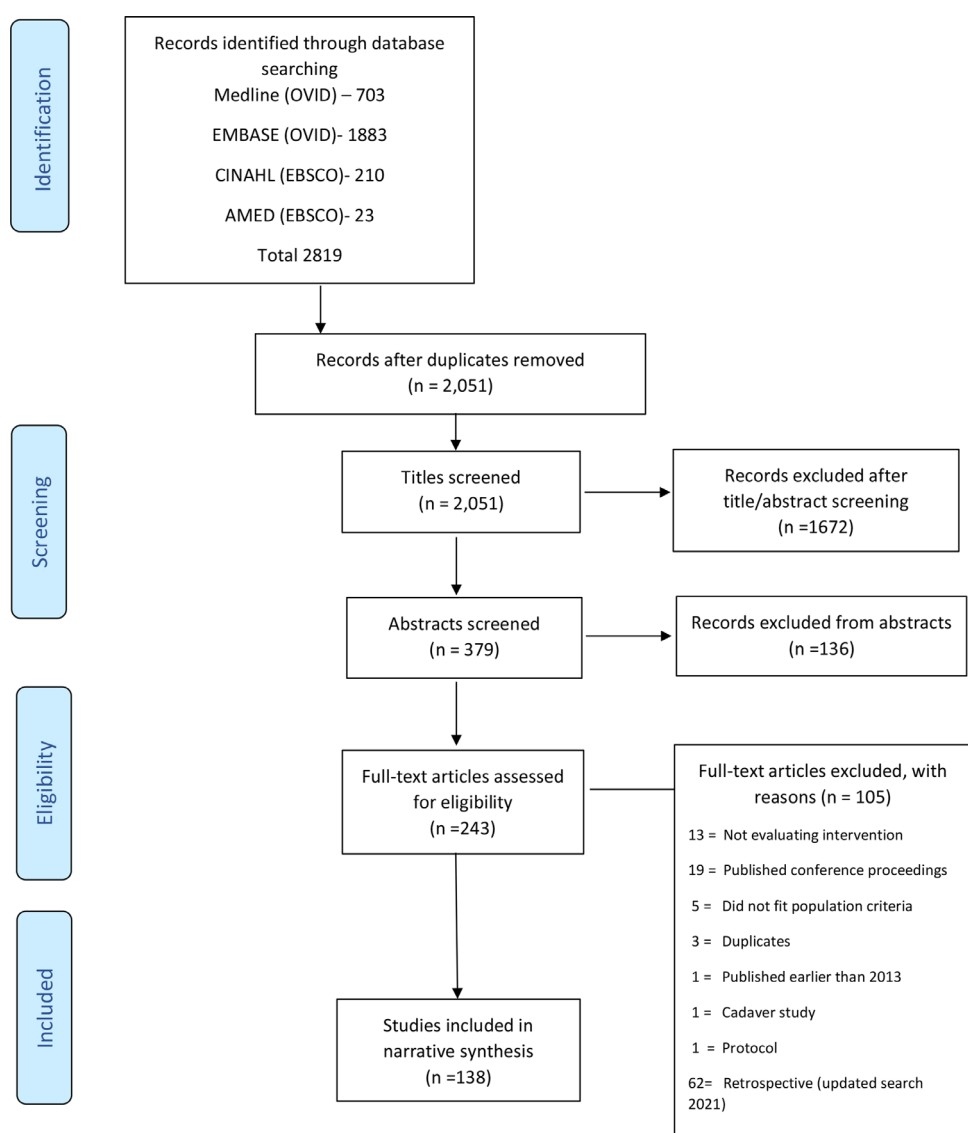

**Figure 1** Preferred Reporting Items for Systematic Reviews and Meta-Analysis flow diagram.

different time points were noted as one outcome. Where outcomes were assessed using an instrument containing several items, each individual item was assigned an outcome name using the International Classification of Functioning and following standard linking rules.[26]

CM categorised all outcomes into an outcome taxonomy developed by COMET for categorising outcomes for COS development.[27] These included 5 core areas and 38 outcome domains. This is presented in online supplemental file 4. A long list of all categorised outcomes was presented to researchers (JC and CJH) at a face-to-face meeting where the categorisation of all outcomes was reviewed using the recommended taxonomy. Subdomains were created within the larger taxonomy to manage the large variation in TBPI clinical outcomes extracted. Disagreements not resolved at this stage were discussed further with subject experts (eg, the adverse event domain was discussed with a surgeon).

Owing to the diversity in terminology used to report outcomes, we grouped similar outcomes within each subdomain. It is recommended that outcomes with different words, phrasing or spelling addressing the same concept should be categorised as a unique outcome.[28] For example, active range of motion of shoulder abduction and active goniometry of shoulder abduction were named as active shoulder abduction range and grasp strength and grip strength were named as grip strength. Independent meetings were held with four subject experts to ratify and define unique outcome names within each domain.

### Patient and public involvement
The need for a COS in TBPI care was conceived following discussions with patients and health professionals. Patients highlighted the diverse effect the injury has on their life and that often these outcomes were overlooked by professionals, such as body image. There is a patient advisory group for the COS and the systematic review was discussed at these meetings. Patients were not actively involved in data collection or analysis of this review. Dissemination will occur at the annual traumatic brachial plexus charity UK meeting where updates from the project are presented yearly and through a 6-monthly newsletter.

### RESULTS
#### Included studies
The search retrieved 2819 studies, after removing duplicates 2051 studies remained.

Title and abstract review identified 243 potentially relevant articles. Of these, 105 studies did not meet the inclusion criteria and were excluded (PRISMA flow diagram; figure 1); thus, 138 studies formed the basis of this review. All included studies are presented in online supplemental file 5.

#### Study characteristics
Thirty-three countries from six continents recruited 3328 participants into the 138 studies (table 1). Of the

**Table 1** Characteristics and demographics of included studies

|  | Study number (%) |
|---|---|
| Number of retrospective studies | 87/138 (63) |
| Number of prospective studies | 24/138 (17) |
| Number of case studies | 23/138 (17) |
| Randomised controlled trial | 4/138 (3) |
| World region recruitment |  |
| Asia | 62/138 (45) |
| North America | 20/138 (14) |
| South America | 23/138 (17) |
| Europe | 28/138 (20) |
| Africa | 3/138 (2.2) |
| Australasia | 2/138 (1.5) |
| Year published |  |
| 2013 | 25/138 (18) |
| 2014 | 24/138 (17) |
| 2015 | 15/138 (11) |
| 2016 | 30/138 (22) |
| 2017 | 27/138 (20) |
| 2018 | 11/138 (8) |
| 2019 (prospective only) | 3/138 (2.2) |
| 2020 (prospective only) | 2/138 (1.5) |
| 2021 (prospective only) | 1/138 (0.7) |
| Gender (total 3328) |  |
| Male | 2737/3328 (82) |
| Female | 335/3328 (10) |
| Not stated | 256/3328 (7.7) |
| Site of plexus injury per study (n=138) |  |
| Upper trunk | 27/138 (20) |
| Lower trunk | 10/138 (7.2) |
| Pan plexus (all avulsed) | 52/138 (38) |
| Infraclavicular | 7/138 (5) |
| Mixture | 35/138 (25) |
| Unclear | 7/138 (5) |
| Interventions (n=138) |  |
| Surgical | 118/138 (86) |
| Electrotherapy | 3/138 (2.2) |
| Pain treatments | 11/138 (8) |
| Rehabilitation | 4/138 (2.9) |
| Orthotic | 1/138 (0.7) |
| Stem cell | 1/138 (0.7) |
| Types of surgical intervention (n=118) |  |
| Neurotisation | 66/118 (56) |
| Tendon transfer | 8/118 (6.8) |
| Free flap | 17/118 (14) |
| Multiple surgeries | 12/118 (10) |
| Contralateral C7 | 8/118 (6.8) |
| Other | 7/118 (5.9) |

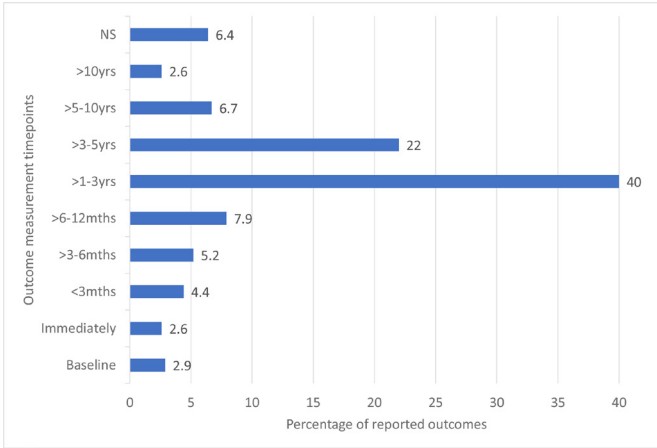

**Figure 2** Outcome measurement timepoints.

138 studies, 87 (63%) were retrospective case series with most studies published from Asia (n=62, 45%). The most frequently studied surgical intervention was neneuritisation (n=66, 56%).

## Outcomes

Extraction of each verbatim outcome domain from each study (eg, range of movement and muscle strength) and those extracted from measures composed of several items identified a total of 1491 verbatim outcomes. After removing duplicates, 157 different unique outcomes remained. No single outcome was reported across all 138 studies.

### Outcome definition variation

Many outcomes were not clearly defined and different terms were frequently found for the same concept. For example, shoulder abduction strength was described in 11 different ways including 'deltoid strength', 'motor function of axillary nerve', 'motor recovery of shoulder abductors', 'muscle power supraspinatus', 'motor function of deltoid' and 'motor function of supraspinatus'.

### Outcome timing variation

Forty per cent of outcomes were measured between 1 and 3 years following intervention. For >6% of outcomes, the timing of the measurement was not stated (see figure 2).

### Outcome domains

The 157 different types of outcomes were categorised into 4 core areas (physiological and clinical, life impact, resource use, adverse events/complications) and 11 domains according to the COMET recommendations[24] (see online supplemental file 6). The core area physiological/clinical included three domains: musculoskeletal and connective tissue outcomes, nervous system outcomes and general/symptom outcomes. The core area, life impact, included seven domains: physical functioning, social functioning, role functioning, emotional functioning, global quality of life, perceived health status and delivery of care. The core area, resource use, included one domain: hospital resources. The core area, adverse events, included one domain: adverse events. No outcome could be placed into the core area, death.

Tables 2–4 summarise the number of unique outcomes within each domain and the number of studies reporting these outcomes in each core area. The most frequently reported domains were all in the physiological/clinical core area and included musculoskeletal and connective tissue (86%), nervous system (33%) and symptoms (38%). Forty-six studies (33%) reported complications/adverse events.

### Outcome measurement

In addition to extraction of standalone clinician-reported outcomes and PROs such as muscle power, range or movement or return to work, outcomes were also extracted from individual items contained in a total of 32 different instruments; PRO measures (n=22), combined clinician-reported and patient-reported measures (n=3) and performance measures (n=7), see table 5. These measures were reported 98 times in the included publications. Most outcome measures were used once (n=22/32, 69%). The most frequently reported measures were the Disabilities of the Arm Shoulder and Hand (DASH[29]) questionnaire (n=28 studies, 29%) and the Visual Analogue Scale (n=20, 20%). The median number of items per instrument was 15 ranging from 1 (Visual Analogue Scale, Numerical Rating Scale and Wong Baker Faces Rating

| Table 2 | Physiological/clinical core area | | |
|---|---|---|---|
| **Outcome domains** | **Number of unique outcomes reported within domain** | **Examples of unique outcomes** | **Number of studies reporting outcomes in domain (%)** |
| Musculoskeletal and connective tissue | 18 | Active range of movement, muscle strength, muscle fatigue | 119/138 (86) |
| Nervous system | 15 | Progression of nerve regeneration, ability to feel light touch, ability to feel pain | 46/138 (33) |
| General/ symptoms | 23 | Pain intensity/relief, pain duration, pain quality, pain when arm exposed to cold, stiffness, sleep, paresthesia | 52/138 (38) |

**Table 3** Life impact core area

| Outcome domains | Number of unique outcomes reported within domain | Examples of unique outcomes | Number of studies reporting outcomes within domain (%) |
|---|---|---|---|
| Physical functioning | 19 | Reaching, fine hand movement | 35/138 (25) |
| Role functioning | 23 | Return to work, impact on normal hobbies | 38/138 (27) |
| Social functioning | 7 | Social activities with family | 32/138 (23) |
| Emotional functioning | 13 | Body image, acceptance | 34/138 (25) |
| Global quality of life | 1 | Quality of life | 2/138 (1.5) |
| Perceived health status | 1 | Health status rating | 9/138 (6) |
| Delivery of care | 13 | Patient satisfaction, quality of care, patient preference, time to surgery | 11/138 (8) |

Scale)[30] to 54.[31] These items mapped to 34 different outcome domains.

There was wide variation in the methods used to measure outcomes. This is presented in online supplemental file 7.

For example, 63 different measurements were used to evaluate muscle function, including the British Medical Research Council,[32] 12 different modifications of the British Medical Council, Isokinetics, Dynamometry and Constant-Murley score.[33] In addition, it was often not clear which instrument was used for measurement of the outcomes. For example, the instrument used to measure active range of movement was not reported in 34% of total times (63/186) the outcome was assessed. Finally, with regards to method of measurement, 61 studies employed a PRO instrument to evaluate the intervention. Prospective and randomised controlled trials were more likely to evaluate outcomes with a PRO (58%;15/26) compared with 36% (31/87) of retrospective studies.

### Outcome reporting bias

Figure 3 illustrates the reporting status of outcomes (n=173) across the included prospective case series, cohort and randomised controlled studies (n=26). Fewer than one-third of the outcomes in the prospective case series and cohort studies and half of outcomes in randomised controlled studies were 'completely' reported.

### DISCUSSION

This systematic review aimed to identify what outcome domains have been reported in studies evaluating interventions for TBPI, examine outcome definitions and timepoints and identify the instruments used to assess outcomes. We found a wide variation in the reported outcomes, timing of outcomes and outcome instruments used. Furthermore, a lack of standardised definition for commonly reported outcomes was observed. This heterogeneity in outcome reporting across studies hinders evidence synthesis and results in research waste.[34]

**Table 4** Adverse events and resource use core areas

| Outcome domains | Number of unique outcomes reported within domain | Examples of unique outcomes | Number of studies reporting outcomes within domain (%) |
|---|---|---|---|
| Adverse events core area | | | |
| Donor site morbidity | 3 | Motor weakness, sensory loss | 24/138 (17) |
| Musculoskeletal | 7 | Co-contraction, passive movement | 12/138 (8.7) |
| Respiratory | 4 | Pneumothorax | 6/138 (4.4) |
| Vascular | 7 | Haematoma | 7/138 (5.1) |
| Infection | 1 | Infection | 3/138 (2.2) |
| General non-specified complications | 1 | General complications | 3/138 (2.2) |
| Resource use core area | | | |
| Hospital resource use | 1 | Operation time | 1/138 (.7) |

**Table 5** Outcome measures used in included studies

| | | Number of items | Number of scales | Frequency (n=98) |
|---|---|---|---|---|
| PRO measures | Upper limb physical function measures (n=17) | | | |
| | DASH | 38 | 3 | 28 |
| | Quick DASH | 19 | 3 | 1 |
| | Upper Extremity Functional Index | 20 | 0 | 2 |
| | American Shoulder and Elbow Score | 15 | 0 | 1 |
| | Modified American Shoulder and Elbow Score | 13 | 0 | 1 |
| | Simple Shoulder Test | 12 | 0 | 1 |
| | Michigan Hand Questionnaire | 37 | 0 | 1 |
| Combined Patient and Clinician Reported measures | University of California Los Angeles Shoulder Score | 5 | 0 | 1 |
| | Constant-Murley | 5 | 0 | 1 |
| | Mayo Performance Index | 4 | 0 | 1 |
| Performance measures | Jebsen Taylor | 7 | 0 | 1 |
| | University of New Brunswick Test of Prosthetic Function for Unilateral Amputees | 30 | 3 | 1 |
| | Upper Limb Module Questionnaire | 22 | 3 | 1 |
| | Action Reach Arm Test | 19 | 4 | 2 |
| | Southampton Hand Assessment Procedure | 26 | 0 | 2 |
| | Purdue Peg test | 3 | 0 | 1 |
| | Activities Measure for Upper Limb Amputees | 24 | 0 | 1 |
| PRO measures | Generic questionnaires (n=3) | | | |
| | 36-item short-form survey | 36 | 8 | 8 |
| | Patient Specific Functional Score | 4 | 0 | 2 |
| | EuroQol-5 Dimension (EQ5D) | 6 | 0 | 1 |
| | Condition-specific questionnaires (n=1) | | | |
| | Trinity Amputation and Prosthesis Scale | 54 | 5 | 1 |
| | Symptom-specific questionnaires (n=10) | | | |
| | Visual Analogue Scale | 1 | 0 | 20 |
| | Numerical Rating Scale | 1 | 0 | 6 |
| | Wong Baker Faces Rating Scale | 1 | 0 | 1 |
| | Brief Pain Inventory | 15 | 6 | 4 |
| | Neuropathic Pain Symptom Inventory | 10 | 5 | 1 |
| | University of Washington Neuropathic Score | 10 | 3 | 1 |
| | McGill Pain Questionnaire | 28 | 3 | 2 |
| | McGill Pain Questionnaire Short-Form | 17 | 3 | 1 |
| | McGill Pain Questionnaire (Japanese version) | 17 | 3 | 1 |
| | Self-rating Anxiety Scale | 20 | 0 | 1 |
| | Zung Self-rating Depression Scale | 20 | 0 | 1 |

DASH, Disabilities of Arm Shoulder and Hand; PRO, patient-related outcome.

The most commonly reported core area was physiological/clinical including musculoskeletal, nervous system and symptom domains. Eighty-six per cent of the studies reported musculoskeletal outcomes. However, there were 21 different outcomes reported in this category making comparison between studies difficult. Furthermore, the diversity of measures used to assess the outcomes increases the difficulty with synthesis. For example, muscle function/strength was assessed using 59 different measures, while 10 studies did not report what measure they used. To compound this, muscle strength was assessed by both physical examination by a clinician (86%) and also by asking the patient (10%).

Only 44% of studies (61/138) evaluated PROs and within these studies there was significant heterogeneity in the measurement instrument used. Twenty-five different

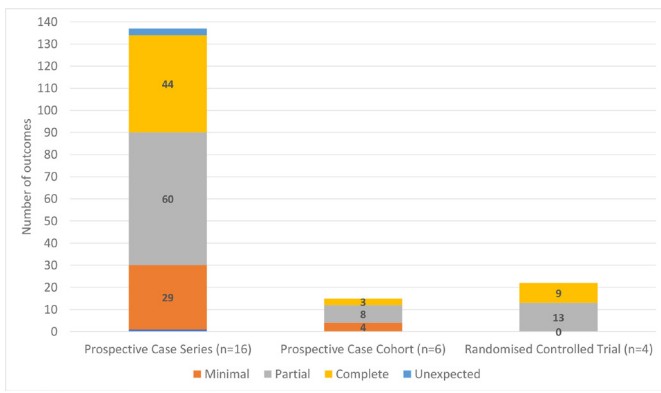

**Figure 3** Cumulative bar chart showing number of outcomes within each reporting bias category across study types.

instruments were used with only 17 ever-used once. The DASH was the most common instrument employed, in almost half the studies evaluating a PRO. The PRO instruments also varied greatly in terms of content with some as simple as a single item while others included up to 54 items. Over 408 individual questionnaire items were evident from the 25 PRO instruments mapping to 34 different outcomes domains. This highlighted a lack of consistency with no domain being measured by all PRO instruments. None of the included PRO assessments were designed specifically for individuals with a TBPI. Although this may be beneficial in terms of comparison with other conditions, such instruments may not be sensitive to issues of importance to patients with TBPI. Finally, it was evident that prospective studies and randomised controlled trials were more likely to use PROs to evaluate interventions. This may correspond with the higher methodological rigour associated with these study designs. However, the majority of studies evaluating interventions in TBPI were retrospective (63%). These issues combined pose major questions regarding the clinical interpretation of results from TBPI studies.

It is clear that that individuals with a TBPI suffer significant emotional and psychosocial issues.[1 35] However, such issues were infrequently and inconsistently measured within this review. Only two studies evaluated quality of life.[36 37] Similarly, physical, role and social functioning outcomes were reported in 25%, 27% and 23% of studies, respectively. This relates strongly to the use of the DASH within the studies. Indeed, emotional functioning was reported in 34 studies, 28 of these studies used the DASH which has one item on confidence and capability mapping to this domain. If the DASH was excluded, only six studies would assess outcomes within the emotional functioning domain. This is surprising considering the existing literature which evidences the complex emotional and psychological factors, individuals face when adjusting to their injury.[1 38]

Complications/adverse events were reported in one-third of the studies. Documentation of complications is crucial to improve patient care and gather data for

benchmarking. In 1992, the Clavien-Dindo classification[39] was introduced to assist with classification of complications to enable comparison between studies.[39] However, within the adverse events outcomes identified in this review there was heterogeneity. Of the 37 verbatim outcomes reported within the donor morbidity (motor) outcome, 19 did not define how this was assessed.

### Outcome reporting bias

Only four studies included in this review were randomised controlled trials.[40–43] However, despite prospective trial registration on a public registry being a condition of publication,[44] none of the randomised trials on TBPI were registered. We also found marked selective outcome reporting in the included prospective and randomised TBPI studies. Most outcomes were only partially reported, frequently lacking specific detail about the outcome result or time of measurement, omitting certain outcome results or lacking detail needed for meta-analysis. This ORB identified in the current TBPI literature threatens the validity of the evidence-based practice in TBPI because it potentially overestimates the effect of treatments or distorts results of studies. This contributes to research waste and critically delays advancement of care for patients.

There are some limitations in this review. We excluded outcomes from older studies to ensure we identified outcomes relevant to contemporary TBPI care. Detailed risk of bias assessment was not undertaken, however the review was designed to identify the breadth of reporting in the literature and not to examine the effectiveness of interventions. The strengths of this review are that the protocol and the data extraction form were prespecified, prospectively registered and the literature search systematic. To account for multidisciplinary perspectives, researchers and clinicians where involved in categorising outcomes into domains. It is the first review to detail the extent of outcome heterogeneity in TBPI research using a systematic method. International and non-English publications were included to reduce the risk of selection bias.

Variation in definitions and measurement of outcomes has been found within other areas of healthcare. Outcome heterogeneity is found in the reporting of outcomes relating to burn care,[45] breast reconstruction[46] and spinal cord injury,[47] among others. A recent review of outcome reporting within burns illustrated wound healing was defined in 166 different ways across 147 studies.[45] A solution to the variation in outcome reporting across studies in TBPI is the development of a COS.[20] This has been shown to improve consistency of outcome reporting.[19 48] Development of a COS in TBPI would not restrict the range of outcomes that can be measured. Researchers and clinicians would still be free to select additional outcomes but the inclusion of such a COS would facilitate synthesis of evidence.[49 50] While work has begun in obstetric brachial plexus injuries to develop a minimum dataset,[51] there is no COS for TBPI.

Considerable work has been done by the COMET initiative through dissemination of resources for COS

development and support for methodological development. COMET recommends a five-step process to develop a COS: define the scope, assess the need, develop the protocol, determine what to measure and determine how to measure.[52] This systematic review addresses these first two steps for the development of the COS in TBPI care. This review has shown the majority of TBPI studies use only clinician reported outcomes to evaluate interventions. However, they do not adequately capture patients' health-related quality of life,[53] and may underestimate the impact of a condition.[54] Concurrent qualitative work to identify outcomes which are important to individuals with a TBPI has been completed by this group. The next stage involves integration of all potential outcomes from this review and the qualitative work into a long list of domains. Healthcare professionals and patients will be invited to prioritise these outcomes during a three-round international online Delphi process and consensus meeting. This will strengthen the case for uptake of a COS for TBPI as it represents patients' and clinicians' perspectives on what outcomes are important. The final stage will map existing validated measures to the outcome domains in the final COS. A future study will evaluate the psychometric properties of those mapped measurement instruments and identify where new measures need to be developed.

## CONCLUSION

This systematic review has shown that outcome reporting in TBPI care is heterogenous and impairment focused with a lack of standardised definitions for commonly reported outcomes. This makes it difficult to compare and combine data from studies to inform decision-making in clinical practice. The measurement instruments used in the studies were also often not clear, particularly when range of movement was assessed. In future studies, authors need to be clearer with descriptions of outcomes assessed and how they were measured. Less than half the studies in this review evaluated outcomes using PRO measures. Given that TBPI has a significant impact on health-related quality of life, it is recommended that authors of future studies include PROs in future studies. We have identified a list of potentially relevant outcomes and categorised these into a clear taxonomy. This will inform the next stage of developing a COS for TBPI where patients, surgeons and therapists will be involved in a consensus process to decide the final outcomes included in a COS for TBPI.

**Acknowledgements** We thank Colin Shirley for his assistance and guidance categorising neurophysiological outcomes.

**Contributors** CM, CJ-H and JC conceived and designed the review. CM and JO'S reviewed the titles, abstracts and full text papers for eligibility. Authors resolved disagreements by discussion or where necessary CJ-H and DMP offered their view. CM and JO'S were responsible for extracting data and data extraction was verified by CJ-H. CM and JO'S independently reviewed outcome reporting bias. CM, CJ-H and JC categorised outcomes. Categorisation was reviewed and edited by DMP and DK. CM prepared the manuscript. CJ-H, JC, DMP, DK and JOS reviewed and edited the manuscript.

**Funding** The study was supported by the National Institute of Health Research (grant number ICA-CDRF-2017-03-039). The NIHR was not involved in any elements of the study design data collection, data analysis and interpretation, or in writing the manuscript. The views expressed are those of the authors and not necessarily those of the NHS, the NIHR or the Department of Health.

**Competing interests** CM, CJ-H, JC, DMP and JO'S declare no potential conflicts of interest with respect to the research, authorship and publication of this article. DK reports grants from NIHR, grants from Innovate UK, grants from NIHR Birmingham Biomedical Research Centre, grants from NIHR SRMRC at the University of Birmingham and University Hospitals Birmingham NHS Foundation Trust, personal fees from Merck, personal fees from GSK, grants from Macmillan Cancer Support, grants from Kidney research UK, outside the submitted work.

**Patient consent for publication** Not required.

**Ethics approval** Ethical approval was not sought for this study because it was a systematic review and did not involve human or animal subjects.

**Provenance and peer review** Not commissioned; externally peer reviewed.

**Data availability statement** All data relevant to the study are included in the article or uploaded as supplementary information.

**ORCID iDs**
Caroline Miller http://orcid.org/0000-0002-4866-0847
Christina Jerosch-Herold http://orcid.org/0000-0003-0525-1282

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
