## [Reviewer comments · BMJ Open]

ARTICLE DETAILS

TITLE (PROVISIONAL)	Developing a core outcome set for traumatic brachial plexus injuries: a systematic review of outcomes.
AUTHORS	Miller, Caroline; cross, jane; O'Sullivan, Joel; Power, Dominic; Kyte, Derek; Jerosch-Herold, Christina

VERSION 1 – REVIEW

REVIEWER	Coroneos, Christopher McMaster University, Plastic Surgery
REVIEW RETURNED	26-Oct-2020

GENERAL COMMENTS	- Every meta-analysis should include assessment of bias. You may find biased studies have different outcomes. You may wish to eliminate a selection of studies (eg. those that did not follow up for long enough).- There are a number of obvious differences between this methodology and that which was previously published in its protocol. These should be adequately addressed in this publication.
--

REVIEWER	Pons, Christelle Fondation ILDYS
REVIEW RETURNED	18-Nov-2020

GENERAL COMMENTS	The aim of this study is to identify what outcome domains are assessed in studies evaluating surgical and non-surgical treatment for traumatic brachial plexus birth palsy (TBPI). This article is part of a more global approach which aims to define a core outcome set for TBBPI This study is well designed, the methodology is appropriate. Introduction It would be interesting to give some information about the more global approach and explain the steps which have been chosen. Regarding the aims, would it be possible to highlight why the second and third objectives are needed? P5 L22 Would it be possible to add some information regarding the country in which these indirect costs were found? Methodology The last search is quite old. Would it be possible to actualize it? Results Contrarily to the first part of the results, the part "outcome measurement" is not very clear for me. Would it be possible to explain how it is possible to obtain 157 different types of outcomes
---

	and only 30 different instruments? Would it be possible to define “performance measures”? Discussion Is it possible to propose some recommendations for the future studies to help the researchers to make the good choices for the outcomes that will be used? Getting the core outcome set may need a long time and having this type of recommendations might be useful for studies that will be made before the definitive COS. For this objective, a discussion regarding metrological properties of the different outcomes might be useful. The next steps are very interesting, could it be possible to give more details regarding it? Strengths and limitations of this study The key points regard only methodological questions. Key results would be also useful.
--	---

VERSION 1 – AUTHOR RESPONSE

Reviewer: 1

Comments to the Author

1. Every meta-analysis should include assessment of bias. You may find biased studies have different outcomes. You may wish to eliminate a selection of studies (eg. those that did not follow up for long enough).

Author response: Thank you for this comment. The study did not claim to be a meta-analysis. i.e This study did not aim to combine the results of multiple studies addressing the same question. The aim was to identify the outcomes currently being assessed and reported in studies evaluating all interventions in traumatic brachial plexus injuries. We included studies of all quality to ensure that no outcomes would be missed. Also as the aim of the overall COMBINE project, is to develop a Core Outcome Set for use in clinical practice / research, the exclusion of case reports and small case series would risk missing important outcomes measured in clinical practice.

2. There are a number of obvious differences between this methodology and that which was previously published in its protocol. These should be adequately addressed in this publication.

We have reviewed the registered protocol on PROSPERO
https://www.crd.york.ac.uk/prospero/display_record.php?ID=CRD42018109843

There were 2 deviations from the original protocol. We have added a supplementary file where we have reported and justified these deviations. Within the paper we have added a sentence signposting readers to this supplementary file to view the deviations.

The revised text reads as follows on P7 line 20-21

“Deviations from the protocol are reported in supplementary file 1.”

Reviewer: 2

Comments to the Author

The aim of this study is to identify what outcome domains are assessed in studies evaluating surgical and non-surgical treatment for traumatic brachial plexus birth palsy (TBPI). This article is part of a more global approach which aims to define a core outcome set for TBBPI. This study is well designed, the methodology is appropriate.

Introduction

1. It would be interesting to give some information about the more global approach and explain the steps which have been chosen.

Author response: Thank you for this comment. We have now included a paragraph to explain where the systematic review sits within the larger program of research.

The revised text reads as follows on P6 lines 10-16

“COS methodology is continuously being refined and promoted by the Core Outcome Measures in Effectiveness Trials (COMET) initiative [20]. Development of a COS usually begins with identification of a long list of outcomes which is then prioritised through a consensus process. This systematic review sits within the larger global COMBINE project to identify a COS for TBPI. A Delphi study and consensus meeting, informed by data from this systematic review and interviews with people with the injury, will prioritise the final COS for TBPI.”

2. Regarding the aims, would it be possible to highlight why the second and third objectives are needed?

Author response: We have now inserted information on why we needed the second and third objectives. The last two lines in the introduction now put into context why measurement instruments and timing of measurement instruments were identified and documented.

The revised text reads as follows on P6 line 12-17

“As a first step in the development of an international COS for TBPI we conducted a systematic review to identify outcomes reported and measurement instruments used and their timing in the literature. The final step of the global project will match the COS to existing validated measurements and make recommendations on when they should be measured, therefore it was necessary to identify currently used instruments and their timepoints also.”

Page 7, line 5

“3. Identify how the outcomes were measured, that is what validated or non-validated instruments are used.”

3. P5 L22 Would it be possible to add some information regarding the country in which these indirect costs were found?

Author response: We have now included that these costs were found in America.

The revised text reads as follows on P5 line 10-12

“The complex and chronic nature of the injury is associated with significant healthcare costs,[7] in addition to indirect costs estimated at up to \$2.34 million (in 2017 dollars) over the lifetime of a manual labourer in the USA with a TBPI,[8].”

4. Methodology

The last search is quite old. Would it be possible to actualize it?

Author response: This systematic review was conducted as part of a larger program of research and we have now made this clearer in the introduction. The results of this systematic review fed into the subsequent development of a questionnaire for the international Delphi and is almost concluded. Redoing/ updating the systematic review would run the risk of identifying new outcomes which would not have been included in the subsequent work on developing consensus.

5. Results

Contrarily to the first part of the results, the part “outcome measurement” is not very clear for me.

Author response: Thank you for this comment. On reviewing the document we agree that we need to be clearer in this section. We have now added information to put this section into context

The revised text reads as follows on page 20 line 2 -6

“In addition to extraction of standalone clinician reported and patient reported outcomes such as muscle power, range or movement or return to work, outcomes were also extracted from individual items contained in a total of 30 different instruments”

6. Would it be possible to explain how it is possible to obtain 157 different types of outcomes and only 30 different instruments?

Author response: Thank you for this comment. We agree that it needs to be explained where the 157 different outcomes were obtained from. We have now included an extra sentences at the beginning of the results section explaining that the 157 outcomes were extracted from both stand-alone outcomes within each study and also measurement instruments. This also explains that the measurement instruments composed of several items.

The revised text reads as follows on page 15 line 2-4

“Extraction of each verbatim outcome domain from each study (e.g range of movement and muscle strength) and those extracted from measures composed of several items identified a total of 1460 verbatim outcomes”

7. Would it be possible to define “performance measures”?

Author response: Performance measures are now defined within the methods section.

The revised text reads as follows on page 10 line 5-10

“Where a validated PRO or performance outcome measurement was used and composed of multiple items, the following data was extracted by the first author: verbatim name of the instrument, verbatim wording for each individual item. A performance outcome measurement was defined as “A measurement based on a standardized task performed by a patient that is administered and evaluated by an appropriately trained individual or is independently completed” [24].”

8. Discussion

Is it possible to propose some recommendations for the future studies to help the researchers to make the good choices for the outcomes that will be used? Getting the core outcome set may need a long time and having this type of recommendations might be useful for studies that will be made before the definitive COS. For this objective, a discussion regarding metrological properties of the different outcomes might be useful.

Author response: Thank you. We agree that this area of the report could be stronger. We have now added three sentences into the conclusion to provide recommendations for future researchers.

The revised text reads as follows on page 28 line 5-10

“The measurement instruments used in the studies were also often not clear, particularly when range of movement was assessed. In future studies, authors need to be clearer with descriptions of outcomes assessed and how they were measured. Less than half the studies in this review evaluated outcomes using PRO measures. Given that TBPI has a significant impact on health-related quality of life, it is recommended that authors of future studies include PROs in future studies”

Author response: With regards to metrological properties of outcome measurements this is the aim of the last study in this larger body of research. The outcomes from the COS will be mapped to existing outcome measurements. Many of these have already been identified within this systematic review. We will then assess the psychometric properties of those measurement instruments which measure the outcomes in the COS. This has now been added into next steps in the discussion. See response to point 9.

9. The next steps are very interesting, could it be possible to give more details regarding it?

We have now inserted the following explanation of the following steps

Author response: Thank you for your comment. We have now added more detail on the larger project and how this systematic review fits into it at the end of the discussion. We have also included when we will examine psychometric properties of the measurement instruments.

The revised text reads as follows on page 28 line 22-24 and page 29 line 1- 6

“The next stage involves integration of all potential outcomes from this review and the qualitative work into a long list of domains. Healthcare professionals and patients will be invited to prioritize these outcomes during a three round international online Delphi process and consensus meeting. This will strengthen the case for uptake of a COS for TBPI as it represents patients’ and clinicians’ perspectives on what outcomes are important. The final stage will map existing validated measures to the outcome domains in the final COS. A future study will evaluate the psychometric properties of those mapped measurement instruments and identify if new measures need to be developed.”

10. Strengths and limitations of this study

The key points regard only methodological questions.
Key results would be also useful.

Author response: Thank you for this comment regarding the strength and limitations section. The author guidelines indicated that this should be about strength and limitations of the study but we are willing to expand if the editor is in agreement with this suggestion.

Reviewer: 1
Competing interests 1: None

Reviewer: 2
Competing interests 1: non declared

VERSION 2 – REVIEW

REVIEWER	Coroneos, Christopher McMaster University, Plastic Surgery
REVIEW RETURNED	29-Dec-2020

GENERAL COMMENTS	 - A number of the comments from both reviewers have not been directly addressed, albeit they were explained. - Basically this was the review of a next step of research, and authors are unwilling to alter it at this point. This is understandable from their end, but doesn't mean that it's perfect. By itself, this paper isn't interesting or novel. It adds little to the literature beyond summarizing what exists. - An assessment of quality is a basic point of any systematic review, and would provide insight to a number of the conclusions you make and are uncertain of. Looking at it from high versus low quality you may see patterns. Trials or cohort studies will likely use more PROMs, case series of novel approaches or techniques will focus on physical outcomes. - All of the points identified by reviewers and not addressed should be added to the limitations paragraph. I.e. there was no quality assessments performed, the search strategy is outdated, etc.
--

VERSION 2 – AUTHOR RESPONSE

Reviewer: 1

Comments to the Author:

- 1. A number of the comments from both reviewers have not been directly addressed, albeit they were explained. Basically this was the review of a next step of research, and authors are unwilling to alter it at this point. This is understandable from their end, but doesn't mean that it's perfect. By itself, this paper isn't interesting or novel. It adds little to the literature beyond summarizing what exists**

Authors response: We have taken on board all your comments about the review. We have now updated the literature search as of 07/May 2021. We have added in the extra studies to the review numbers and the Prisma diagram. All outcomes have been extracted from the new studies including extraction of outcomes from the new measurement instruments identified.

We have undertaken a quality assessment of the studies using a risk of outcome reporting bias tool (detailed below) as well as reviewed the association between study design and the use of PROMS (see below for more detail).

- 2. An assessment of quality is a basic point of any systematic review, and would provide insight to a number of the conclusions you make and are uncertain of. Looking at it from high versus low quality you may see patterns. Trials or cohort studies will likely use more PROMs, case series of novel approaches or techniques will focus on physical outcomes.**

Authors response: Thank you for this suggestion. We have now reviewed and categorised study designs in relation to the use of PROMS. Indeed we did find that higher quality studies were more likely to use PROMS.

The revised text in the results section reads as follows on line 1-3, page 21

“Prospective and randomized controlled trials were more likely to evaluate outcomes with a PRO (58%;15/26) compared to 36% (31/87) of retrospective studies.”

The revised text in the discussion section reads as follows on line 18-24, page 25

“Finally, it was evident that prospective studies and randomised controlled trials were more likely to use patient reported outcomes to evaluate interventions. This may correspond with the higher methodological rigour associated with these study designs. However the majority of studies evaluating interventions in TBPI are retrospective (63%). These issues combined pose major questions regarding the clinical interpretation of results from TBPI studies”

Additionally, to assess quality, we have also conducted a review of outcome reporting bias in all included prospective case series, cohort and randomised controlled trials. New sections in the methods, results and discussion area have been written to explain how the quality assessment was undertaken, what the results were and implications of the quality assessment in the discussion area.

The revised text in the methods section reads as follows on page 9, lines 18-24

“The aim of the study was to identify outcomes reported in studies rather than synthesise data on intervention effectiveness. However, selective outcome reporting can provide information on what outcomes authors prioritize. We used a modified version of Kirkham et al’s matrix(Kirham et al 2018, Deshmukh et al 2021) to assess outcome reporting bias (ORB) in included prospective and RCT studies(See Outcome Reporting Bias instrument in Supplementary Information X). Two independent reviewers (XX &XX) performed the assessment of ORB for all outcomes”

The revised text in the results section reads as follows on page 23 , lines 4-11

Outcome Reporting Bias

“Figure 2 illustrates the reporting status of outcomes across the included prospective case series, cohort and randomized controlled studies (n=26). None of the studies were prospectively registered. Fewer than one third of the outcomes in the prospective case series and cohort studies and half of outcomes in randomized controlled studies were “completely” reported.”

A new figure has been developed and added to the submission to illustrate the extent of the outcome reporting bias.

The revised text in the discussion section reads as follows on lines 1-10, page 27

Outcome Reporting Bias

“Only four studies included in this review were randomised controlled trials (ref, ref, ref, ref). However despite prospective trial registration on a public registry being a condition of publication (ref) none of the randomised studies on TBPI were registered. We also found marked selective outcome reporting in the included prospective and randomised TBPI studies. Most outcomes were only partially reported, frequently lacking specific detail about the outcome result, time of measurement, omitting certain outcome results or lacking detail needed for meta-analysis. This outcome reporting bias identified in current TBPI literature threatens the validity of the evidence base, contributes to research waste and critically delays advancement of care for patients”

All of the points identified by reviewers and not addressed should be added to the limitations paragraph. Ie. there was no quality assessments performed, the search strategy is outdated, etc.

Authors response: We hope that we have addressed most of the limitations which you have identified here. We have now updated the literature search and included the new studies and extracted all outcomes including outcomes from new outcome measurement instruments which were not reported when the review was originally conducted. We have also completed a quality assessment on selective outcome reporting and reviewed the association between the study design and the used of certain outcomes. We believe that this has made the paper much stronger and look forward to your response.

VERSION 3 – REVIEW

REVIEWER	Coroneos, Christopher McMaster University, Plastic Surgery
REVIEW RETURNED	15-Jun-2021
GENERAL COMMENTS	Concerns have been addressed